# Polysulfone Membranes Based Hybrid Nanocomposites for the Adsorptive Removal of Hg(II) Ions

**DOI:** 10.3390/polym13162792

**Published:** 2021-08-19

**Authors:** Abeer M. Alosaimi

**Affiliations:** Department of Chemistry, College of Science, Taif University, P.O. Box 11099, Taif 21944, Saudi Arabia; a.alosaimi@tu.edu.sa

**Keywords:** hybrid nanocomposites, polysulfone, clay, graphene oxide, carbon nanotube, heavy metal removal

## Abstract

Organic–inorganic nanoparticles, which can improve and modify the mechanical and chemical properties of polymers, have been used as fillers to prepare high-performance hybrid nanocomposite membranes. In this study, we explored whether the incorporation of organic nanofillers (graphene (G), graphene oxide (GO), carbon nanotubes (CNTs), or oxidized carbon nanotubes (CNTOxi)) into polysulfone (PSF) and montmorillonite (MMt)-modified PSF membranes could enhance membrane performance for the removal of heavy metal ions from contaminated solutions. These hybrid membranes were prepared by a phase inversion method using chloroform as the solvent. The surface morphologies of the membranes revealed good dispersibility of the organoclay and carbon nanomaterials in the PSF matrix. The hybrid nanocomposite membranes showed significantly improved thermal stability and mechanical properties as compared to the pristine PSF and PSF/MMt membranes. The adsorption efficiencies of these hybrid adsorptive membranes for Hg(II), Pb(II), Sr(II), Fe(III), Zn(II), Ni(II), Al(III), Co(II), Y(III), and Cr(III) were investigated. The PSF/MMt/CNTOxi and PSF/MMt/GO membranes exhibited the highest adsorption efficiencies. In particular, these adsorptive membranes showed selectivity toward Hg(II), and the Hg(II) extraction percentage was maximized at pH 2. The maximum Hg(II) adsorption capacities of PSF/MMt/CNTOxi and PSF/MMt/GO were 151.36 and 144.89 mg/g, respectively, and the adsorption isotherm was in approval with the Langmuir model. These hybrid nanocomposites can be used in water purification application.

## 1. Introduction

In recent years, membrane separation technology has been developed for use in advanced water treatment processes and has played an important role in reducing water pollution [1,2,3,4]. However, water pollution remains an issue because of the continued growth in the number of new organic compounds used in commercial products such as pharmaceuticals, personal care products, and pesticides, as well as the use of toxic heavy metals that can affect human health [5,6,7]. As a result, the availability of clean water for daily life and for agricultural and industrial use has become a global challenge. Moreover, owing to the huge diversity of pollutants, many conventional membrane materials for wastewater treatment are no longer feasible, and the development of new approaches is an ongoing challenge in the field of membrane technology.

Polymeric membranes are commonly used for separation technology owing to their good physicochemical properties, environmental friendliness, effectiveness, and low fabrication costs. In particular, remarkable progress has been made in the field of heavy metal removal from industrial wastewater using polymeric nanocomposite membranes, and considerable research efforts have been focused on the synthesis of membranes with high efficiencies for water purification [8,9]. Polysulfone (PSF) is one of the most popular polymers for use in ultrafiltration membranes because of its high mechanical strength, excellent thermal stability, and superior chemical resistance over the entire pH range [10,11,12]. However, PSF has a lower surface energy and higher hydrophobicity than other hydrophilic polymer membranes, which hinders the performance of PSF membranes in treating wastewater and other pollutants. Numerous efforts have been made to enhance the hydrophilicity of PSF membranes and overcome the serious fouling issue, including surface modification, coating, and hydrophilic polymer blending.

Recently, organic–inorganic materials have been used to prepare hybrid membranes with excellent properties and a wide range of applications [13,14]. For example, Wu et al. prepared a SiO_2_–graphene oxide (GO) nanohybrid/PSF membrane to improve water permeability [15]. Chai et al. embedded Fe_3_O_4_/GO in PSF to enhance permeability and humic acid rejection [16]. To date, many inorganic nanoparticles have been incorporated in the fabrication of membranes, including metal oxides, zeolites, and clay [17,18]. In particular, clay has attracted recent interest as a filler in composites because of its low cost, environmental friendliness, and abundance. Polymer–clay nanocomposites have been demonstrated to have various industrial, technological, and electronic applications, and have been widely used for water purification because clay can enhance the properties of polymer matrices, even when added in small amounts, such as the hydrophilicity property, and improves the efficiency of the polymer matrix for pollutant removal in aqueous systems [19,20,21,22].

Among the many types of clay used in polymer–clay nanocomposites, montmorillonite (MMt) is the most common clay mineral for the preparation of filtration membrane nanocomposites. Many studies have demonstrated that the addition of MMt improves the properties of filtration membranes, including the surface hydrophilicity, water permeability, pore structure, and thermal and mechanical properties [23]. However, similar to other clays, MMt is naturally hydrophilic, hindering its interactions with and distribution in most types of polymers, which are typically hydrophobic. To improve compatibility, MMt can be treated with organic cations, called surfactants, before being mixed with a polymer to form a nanocomposite membrane. The ion exchange method is a traditional and simple method for modifying layered silicate surfaces, in which cations between the silicate layers are replaced with organic cations [24], thus allowing MMt to be compatibilized with a polymer matrix. However, the incorporation of inorganic nanoparticles into polymer matrices only results in a limited improvement in properties. Therefore, the introduction of carbon nanomaterials such as GO, carbon nanotubes (CNTs), and C_60_ to produce novel nanohybrid materials has also been investigated.

Compared with graphene (G) and other carbon nanomaterials, GO is attractive as a nanofiller for the preparation of nanohybrid membranes because of its unique properties, such as a high surface area and excellent physical properties [25,26,27,28]. Further, different forms and hybrid/composite structures of GO can be designed [29]. The surface of GO has hydrophilic oxygen-containing functional groups that can adsorb organic species via various mechanisms, including electrostatic interactions and ion exchange [30,31]. For example, GO was introduced into a PSF ultrafiltration membrane to improve hydrophilicity and antifouling abilities of the membrane for water treatments as adsorbent of a mixture of selected organic contaminants of environmental relevance [32]. In addition, a GO-based polymer membrane was used as a filtration or adsorptive membrane for Hg separation from wastewater [33].

CNT–polymer nanocomposites have also attracted attention for many applications, including membrane technology, because CNTs possess excellent physicochemical properties that can improve the performance of polymeric materials [34]. Additionally, CNTs have been used as nanosorbents for water treatment because of their high adsorption capacities for organics [35]. Xu et al. embedded raw CNTs and oxide-functionalized CNTs in PSF to produce nanocomposite membranes with enhanced properties and filtration performance [36]. Further, pristine CNTs can be functionalized to enhance the chemical reactivity of the surface, and the effect of CNT carboxylation on the properties of nanocomposite polymer membranes has been investigated [3,37,38,39].

In this study, we investigated the preparation of hybrid nanocomposites, in which both MMt and carbon nanomaterials were incorporated as high-performance polymer adsorptive membranes. PSF hybrid nanocomposites were prepared using the phase inversion method. Organic nanofillers (G, GO, CNTs, or oxidized carbon nanotubes (CNTOxi)) were added to MMt-modified PSF in fixed proportions. The morphology, mechanical properties, and thermal stability of the composite membranes were evaluated to obtain a better understanding of the effects of MMt and G, GO, CNTs, or CNTOxi on the physicochemical properties of the membrane. Finally, the analytical potential of the adsorptive membranes for the selective extraction of several heavy metal ions, especially Hg(II), was explored.

## 2. Experimental Details

### 2.1. Chemicals and Reagents

Polysulfone pellets having a molecular weight of 60,000 were purchased by ACROS ORGANICS CO. Carbone materials (G, GO and MWCNT) were purchased from Nanotechnology CO. LTD Egypt. Montmorillonite-modified nanoclay by 25–30% trimethylstearylammonium salt. Chloroform, nitric acid, sulfuric acid, and stock standard solutions of 1000 mg L^−1^ Sr(II), Co(II), Zn(II), Pb(II), Al(III), Y(III), Cr(III), Fe(III), Ni(II) and Hg(II)were purchased from Sigma-Aldrich (Milwaukee, WI, USA) and were used without any further purifications.

### 2.2. Acid Treatment of CNTs

The carboxylation treatment by acid oxidization [40]: 5 mg MWCNT was added into 6.5 mL a mixture of nitric acid/sulfuric acid (1:3 in volume) and refluxed at 80 °C for 3 h. Then the mixture was introduced in an ultrasonic bath for 2 h at ambient condition. After dilution with DI water, the MWNTs were filtrated through a 0.45 μm Millipore nylon filter membrane. Then the product was washed by distilled water until the pH of the filtrate reached near neutral and then dried at 80 °C in a vacuum oven for 24 h.

### 2.3. Preparation of Hybrid Nanocomposite Membranes

Polysulfone/Organoclay/Organic nanofiller (G, GO, CNTs, or CNTOxi) hybrid membranes were fabricated by the phase inversion method. First, PSF (4 g) was dissolved in a suitable amount of chloroform under constant stirring at room temperature to form homogeneous solution. Then, MMt and graphene particles (2.5% wt for each one) were dispersed in chloroform, each separately, and sonicated for 10 min until forming a clear homogenous suspension. Then they were added to the PSF solution. Next, the mixture was stirred for 30 min. After that, this solution was cast onto a glass blade (mold). Finally, the membranes were dried at room temperature for evaporation of CHCl_3_ and yielded homogenous films. The membrane thickness was approximately 25 μm. This procedure was used with each of the following organic nanofillers (GO, CNTs, or CNTOxi). The sample’s composition is shown in Table 1. 

### 2.4. Adsorption Method Procedure

For adsorption tests, we followed the procedure as mentioned in the literature on our samples [7]. Briefly, stock solutions of Hg(II), Pb(II), Sr(II), Fe(III), Zn(II), Ni(II), Al(III), Co(II), Y(III), and Cr(III) were prepared in 18.2 MΩ·cm distilled deionized water and stored in the dark at 4 °C. For selectivity study of all the prepared membranes in this study, standard solutions were prepared with 5 mg/L of each metal ion and the pH of the solutions was adjusted from 1.0 to 9.0 using the appropriate buffer solutions. All standard solutions were individually mixed with 25 mg PSF/MMt/CNTOxi and/or PSF/MMt/GO in order to study the effect of pH on the selectivity of these material adsorptions across Hg^+2^ ion. A mechanical shaker was applied for all mixtures for 1 h at 170 rpm at room temperature. Regarding the study of the adsorption capacity of Hg^+2^ under batch conditions, standard solutions of 5, 10, 15, 20, 25, 50, 100, 125, 175, 200, and 250 mg/L Hg^+2^ were prepared as the above procedure and adjusted to the optimum pH value of 2.0 and individually mixed with 25 mg PSF/MMt/CNTOxi and/or PSF/MMt/GO using a mechanical shaker.

### 2.5. Instrumentation

The membranes were also studied by Fourier transform infrared (FTIR) spectra within the wavelength range 500–4000 cm^−1^ and were obtained using a Bruker Vertex 80v spectrometer at room temperature. The morphology of the membranes was investigated by the scanning electron microscopy (SEM) using an JEOL instrument (JSM-6390LA) with EDX. The membrane samples were sputter coated with Au atoms, and the surface and cross-section of the membranes were studied by SEM at 5.0 kV. XRD patterns were performed for the nanocomposite membranes, which were obtained using Bruker diffractometer (Bruker D8 advance target) under the following conditions: 40 kV–30 mA; Cu Kα radiation (*λ* = 1.54060 Å); and at a rate of 0.6°/min in the range from 5 to 80° (2*θ*). Thermogravimetric (TG) analyses of the membranes were performed using a Shimadzu Thermal Analyzer under a nitrogen atmosphere, in the temperature ranging from the room temperature to 800 °C.

Mechanical response of the materials was studied by using Universal Testing Machine model GTTGS-2000 at a crosshead speed of 1.0 mm/min, according to ASTM D-3039. The size of the sample was 10 cm in length and 1.5 cm in width. Impact test was carried out using Zwick Pendulum Tester Model 5101, charpy method on samples with dimension of 6.5 × 1.2 cm. The pendulum energy of 2 J was used for all the samples. The test was conducted according to ASTM D252. The values of properties were reported based on the average of five measurements for each sample.

Inductively Coupled Plasma Optical Emission Spectrometry (ICP-OES) measurements were obtained by use of a Perkin Elmer ICP-OES model Optima 4100 DV, USA. The ICP-OES spectrometer was used with the following parameters: FR power, 1300 kW; frequency, 27.12 MHz; plasma gas (Ar) flow, 15.0 L/min; auxiliary gas (Ar) flow, 0.2 L/min; nebulizer gas (Ar) flow, 0.8 L/min; nebulizer pressure, 2.4 bar; sample pump flow rate, 1.5 mL/min; integration time, 3 s; and wavelength range of monochromator, 165–460 nm.

## 3. Results and Discussion

### 3.1. Characterization of Hybrid Nanocomposite Membranes

#### 3.1.1. Fourier Transform Infrared (FTIR) Spectroscopy

The dispersion of organic and inorganic particles in the polymer matrix plays an important role in the performance of hybrid membranes and is considered one of the greatest challenges in their development. FTIR spectra of the hybrid membrane nanocomposites were collected from 4000 to 400 cm^−1^, as shown in Figure 1. In the FTIR spectrum of PSF, the peak at 1236 cm^−1^ is attributed to the stretching vibration of C–O–C groups, the peaks at 1480 and 1585 cm^−1^ are associated with the skeletal vibrations of aromatic hydrocarbons, and the peaks at 1154 and 1305 cm^−1^ are assigned to the symmetric and asymmetric stretching of sulfone groups [4,41]. The spectral characteristics of the composite materials were very similar to those of PSF, which indicates that there were no interactions between the PSF chains and the various nanofillers. The FTIR spectrum of PSF/MMt/CNTOxi exhibited a characteristic carbonyl peak at 1743 cm^−1^ corresponding to the carboxylic groups in CNT-COOH and a band at 3639 cm^−1^ corresponding to hydroxyl groups [34]. The FTIR spectrum of the PSF/MMt/GO membrane showed a distinct carbonyl peak at 1738 cm^−1^ and a broad band at 3624 cm^−1^, which suggests that a hydrophilic layer was formed on the surface of the membrane owing to the hydrophilic functional groups of GO [11,28].

#### 3.1.2. X-ray Diffraction (XRD)

XRD was utilized to determine the dispersion of the organic and inorganic nanofillers within the PSF matrix. Figure 2 shows the XRD patterns of pristine PSF and its hybrid nanocomposites. The XRD pattern of PSF showed a broad peak at 17° owing to the amorphous structure of the PSF chains. Various studies have indicated that GO exhibits a diffraction peak at 10.8°, whereas G, CNTs, and CNTOxi exhibit diffraction peaks at 26° [28,42]. The modified MMt exhibits a diffraction peak at 35.9° [43]. From Figure 2, the peaks of MMt that had a higher alkyl ammonium chain as a surfactant disappeared when mixed with PSF and G, GO, CNTs, or CNTOxi. In addition, the G, GO, CNTs, and CNTOxi with PSF had peaks in the same 2 theta values, which indicates no structural distortion of these fillers due to mixing with PSF. Furthermore, changes in the intensity of diffraction peaks indicate variations in the crystallinity of composite materials. As shown in Figure 2, the diffraction peaks in the XRD patterns of PSF/MMt/CNTs and PSF/MMt/CNTOxi had low intensities, indicating a decrease in crystallinity, which suggests that the nanomaterials had good compatibility and were well dispersed within PSF. Conversely, the diffraction peak for G in PSF/MMt/G was sharp. In the PSF/MMT/GO sample, the peak of GO at 10.8° was not observed because it was overlapped by the broad peak of PSF.

#### 3.1.3. Scanning Electron Microscopy (SEM)

As good dispersion of the nanofillers within PSF is essential to maintain the polymer matrix properties, the morphologies of the hybrid nanocomposite membranes were examined using SEM. Figure 3 shows representative SEM images of the top surfaces (Figure 3a–f) and cross-sections (Figure 3g–l) of the membranes. The pristine PSF membrane possessed a smooth and homogenous surface (Figure 3a). The addition of MMt to PSF led to a random and homogeneous distribution of MMt nanoparticles over the entire surface (Figure 3b). As shown in Figure 3c–f, the surface became much rougher with the addition of the carbon-based nanofillers, which were well dispersed in the polymer matrix without any serious agglomeration. The nanofillers appeared pale on the surface owing to the good compatibility between PSF, MMt, and the carbon nanomaterials. Moreover, no cracks were observed on the surface, indicating that the membranes did not become brittle upon the addition of nanofillers. The cross-sectional images of the membranes reveal an asymmetric sponge-like structure (Figure 3g–l). All membranes exhibited a dense skin layer, and the boundaries between the nanofillers and the PSF matrix were not defined, suggesting strong interfacial adhesion. These results confirm good compatibility between the nanofillers and PSF [1,15,29,32,44].

#### 3.1.4. Thermogravimetric Analysis (TGA)

The thermal stabilities of pristine PSF and hybrid nanocomposites were examined by TGA at a rate of 10 °C/min under a nitrogen atmosphere (Figure 4). The TGA curves of all the membranes showed one main degradation step over the temperature range of 70–750 °C, except PSF/MMt started degradation from 35 °C. The degradation before 230 °C was attributed to the removal of water or other impurities; the decomposition step after 230 to 750 °C for pristine PSF and the nanocomposites was due to PSF and nanofillers degradation.

Generally, the thermal stabilities of the nanocomposite membranes were higher than that of the pristine PSF membrane. For PSF/MMt, the addition of MMt did not improve the thermal stability of the membrane, primarily because the organically modified clay contained surfactants with low thermal stability [19]. In the hybrid nanocomposite membranes, the presence of MMt and the carbon nanomaterials improved the thermal stability of PSF. Particularly, PSF/MMt/CNTs demonstrated more thermal stabilization compared to other nanocomposite membranes. Table 2 lists some of the important thermogravimetric data. Degradation temperatures, T10 and T50, were enhanced by 150 °C due to the incorporation of nanofiller in pure PSF. More importantly, the solid residue left at 750 °C are the main criteria indicating the thermal stability of the nanocomposites. The higher these values are, the higher is the thermal stability.

#### 3.1.5. Mechanical Properties

The mechanical performance of polymer nanocomposites depends on various factors such as the dispersion state, the nanofiller characteristics, and interfacial interactions between the matrix and the filler. The mechanical stabilities of pristine PSF, PSF/MMt, PSF/MMt/G, PSF/MMt/GO, PSF/MMt/CNTs, and PSF/MMt/CNTOxi were evaluated at room temperature in the dry state. The flexural modulus of elasticity and tensile strain at break values of the membranes are listed in Table 3, whereas the tensile properties and impact strength are listed in Table 4.

Various studies have reported that the addition of clay enhances the mechanical properties of membranes because of an increased sponge-like structure and smaller voids. However, membranes synthesized with clay and PEG have poor mechanical properties owing to the presence of pores, reduced stiffness, and tensile strength [23]. Other studies have found that the addition of G or CNTs to PSF can also improve the mechanical properties. In this study, the hybrid membranes exhibited excellent mechanical properties because of the presence of both MMt and a carbon nanofiller [30,36].

The hybrid nanocomposite membranes exhibited better flexural properties (Table 3) and tensile properties (Table 4) than the pristine PSF and PSF/MMt membranes. This enhancement was attributed to the excellent dispersion of the binary nanofillers in the PSF matrix and a strong interface resulting from good interactions between the nanoparticles and the polymer chains [3,28]. However, the tensile strain at break values was decreased for the hybrid membranes (Table 3), which indicates that the hybrid membranes are more brittle, likely because the binary fillers hinder the sliding of PSF chains to a greater extent than a single filler. Furthermore, PSF/MMt/GO showed remarkably improved flexural and tensile properties as compared to PSF/MMt/G. Similarly, PSF/MMt/CNTOxi exhibited greater mechanical strength than PSF/MMt/CNTs. This effect was attributed to the improved interfacial compatibility owing to hydrogen bonding between the nanomaterials and the PSF matrix.

### 3.2. Adsorption Performance of Hybrid Nanocomposite Membranes

PSF has been extensively used to remove diverse environmental pollutants. Therefore, the hybrid nanocomposite membranes were used to extract toxic Hg(II) from aqueous solutions as surface-selective adsorbents. All the nanocomposite membranes yielded significantly higher adsorption efficiencies than pristine PSF and PSF/MMt. As PSF/MMt/CNTOxi and PSF/MMt/GO exhibited the highest adsorption efficiencies among the investigated membranes, their adsorption performance was investigated in detail. The distribution coefficient (*K_d_*) for the selectivity of the membranes toward different metal ions was calculated using the following equation:(1)$$K_{d}=\frac{\left(C_{o}-C_{e}\right)\;}{C_{e}}\;\times \;\frac{V}{M}$$
where *C_o_* and *C_e_* are the initial and final concentrations before and after filtration, respectively, *m* is the weight of the adsorbent (g), and *V* is the volume (mL). Table 5 contains the distribution coefficient values for all the metal ions in this study.

As shown in Table 5, among the investigated metal ions, PSF/MMt/CNTOxi had the higher distribution coefficient value for Hg(II) (61,357.25 mL/g). This result indicates that PSF/MMt/CNTOxi is selective toward Hg(II).

#### 3.2.1. Effect of pH

Owing to the effect of H^+^ ions on the ionization degree and adsorbate species, the pH plays a vital role in the extraction of metal ions from aqueous media [7]. Therefore, the effect of pH on the adsorption of Hg(II) by PSF/MMt/CNTOxi and PSF/MMt/GO was investigated. Using 2 mg/L Hg(II), the solution pH was varied from 1 to 11. Each standard solution was mixed with 20 mg of PSF/MMt/CNTOxi or PSF/MMt/GO at 25 °C. As shown in Figure 5 shows, the solution pH had a remarkable effect on the extraction of heavy metals process, with the extraction percentage of Hg(II) first increasing and then decreasing as the pH value increased.

PSF/MMt/CNTOxi and PSF/MMt/GO showed the highest Hg(II) extraction percentages at pH 2.0 (98% and 95%, respectively), which indicates that the selectivity toward Hg(II) was maximized at this pH value. This enhanced Hg(II) extraction performance can be explained by electrostatic interactions between the protonated carbonyl groups of PSF/MMt/CNTOxi or PSF/MMt/GO and the negatively charged species HgCl_4_^−^ at pH 2.0, which is the main form of Hg(II) ion in HCl solution. Thus, Hg(II) was selectively extracted from the solution. Based on these results, the optimum pH value of 2.0 was used in subsequent investigations of the Hg(II) uptake capacities of PSF/MMt/CNTOxi and PSF/MMt/GO under static conditions.

#### 3.2.2. Determination of Adsorption Capacity

The Hg(II) uptake capacity was investigated at pH 2.0 using different amounts of Hg(II) mixed with 20 mg of PSF/MMt/CNTOxi or PSF/MMt/GO using a batch method. From the obtained adsorption isotherms, the adsorption capacities of PSF/MMt/CNTOxi and PSF/MMt/GO for Hg(II) were found to be 151.36 and 144.89 mg/g, respectively, at 150 mg/L of Hg(II) concentration (Figure 6).

Figure 6 also shows that there was a small decrease in the Hg(II) uptake capacities of PSF/MMt/CNTOxi and PSF/MMt/GO after saturation, at higher than 150 mg/L. This behavior was attributed to the binding sites of PSF/MMt/CNTOxi or PSF/MMt/GO being saturated with HgCl_4_^–^ species, especially at 250 mg/L, with the highest concentration of Hg(II). The stabilities of PSF/MMt/CNTOxi and PSF/MMt/GO were investigated over three cycles. The membranes exhibited little change in adsorption capacity, indicating their excellent stability, and thus could be reused with high efficiency [45].

#### 3.2.3. Adsorption Isotherm Models

Adsorption isotherm models can be described by Langmuir adsorption models. The experimental data were fitted well by the Langmuir equation to describe adsorption isotherm models, as illustrated in Figure 7.

According to the Langmuir adsorption isotherm, the uniformity of non-interacting surface area sites was measured by the mathematical expression of this model:*C_e_*/*q_e_* = (*C_e_*/*Q_o_*) + 1/*Q_o_b*(2)
where *C_e_* is the concentration and *q_e_* is the amount of metal ions in solution at equilibrium, respectively. *Q_o_* and *b* are Langmuir constants for the adsorbent, they can be calculated from a linear plot of *C_e_*/*q_e_* against *C_e_*, which has a slope and intercept equal to 1/*Q_o_* and 1/*Q_o_b*, respectively. Moreover, an equilibrium parameter, *R_L_*, is from the essential characteristics of the Langmuir adsorption isotherm model, which is defined as follows:*R_L_* = 1/(1 + *bC_o_*)(3)
where *b* is the Langmuir constant, which indicates the adsorption nature and different isotherm shapes, and *C_o_* is the initial concentration of Hg(II). The value of *R_L_* describes the adsorption isotherm nature, and values of 0 < *R_L_* < represent favorable adsorption. A linear plot was obtained from the Langmuir isotherm equation based on least-squares fitting, confirming the validity of the Langmuir adsorption isotherm model for the adsorption process, as shown in Figure 7. The above results indicate that adsorbed Hg(II) forms a monolayer on the homogeneous PSF/MMt/CNTOxi surface, which is a monolayer during the adsorption process. The Langmuir constants *Q_o_* and *b* were calculated as 149.36 mg/g and 0.36 L/mg, respectively, for PSF/MMt/CNTOxi, and 142.48 mg/g and 0.28 L/mg, respectively, for PSF/MMt/GO (Table 6).

Table 6 shows the data could be satisfactorily fitted with the Langmuir model. The correlation coefficient (*R*_2_) obtained from the Langmuir model was 0.992 for the adsorption of Hg(II) on PSF/MMt/CNTOxi and 0.985 for PSF/MMt/GO. The *R_L_* value for Hg(II) adsorption on PSF/MMt/CNTOxi was 0.04, which indicated a highly favorable adsorption process. In addition, the Hg(II) adsorption capacity (149.36 mg/g) was in good agreement with that (151.36 mg/g) obtained experimentally from the adsorption isotherm.

## 4. Conclusions

In the present study, G, GO, CNTs, or CNTOxi-incorporated PSF/MMt nanocomposite adsorptive membranes were fabricated for the removal of heavy metal ions from aqueous media. FTIR spectroscopy, XRD, and SEM indicated good compatibility and excellent dispersibility of the organoclay (MMt) and carbon nanomaterials (G, GO, CNTs, or CNTOxi) in the PSF matrix. The thermal stabilities of the hybrid nanocomposite membranes were higher than that of pristine PSF and PSF/MMt. Moreover, all the hybrid nanocomposites (especially PSF/MMt/CNTOxi and PSF/MMt/GO) showed enhanced mechanical properties as compared to pristine PSF, proving the formation of a strong interface, which is required for efficient load transfer from the PSF matrix to the binary filler. Furthermore, an investigation of the adsorption performance of the membranes for various metal ions revealed that PSF/MMt/CNTOxi and PSF/MMt/GO had the highest adsorption efficiencies. The adsorption capacities of PSF/MMt/CNTOxi and PSF/MMt/GO for Hg(II) at pH 2 were 151.36 and 144.89 mg/g, respectively, and the adsorption isotherms were in agreement with the Langmuir model. In conclusion, these results confirmed that the PSF membrane by mixing with modified MMt and carbon materials can be a promising material for future industrial applications.

## Figures and Tables

**Figure 1 polymers-13-02792-f001:**
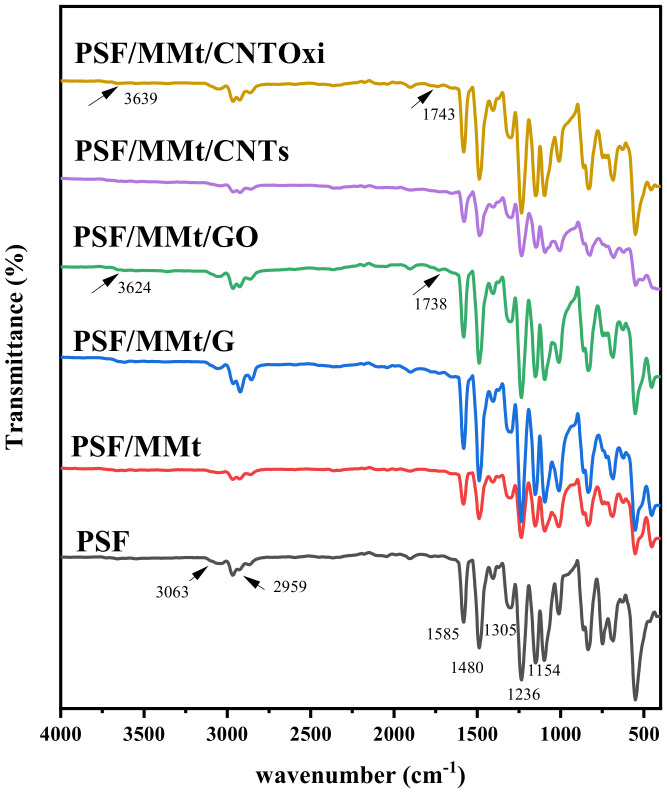
FTIR spectra of pure PSF, PSF/MMt, PSF/MMt/G, PSF/MMt/GO, PSF/MMt/CNTs, and PSF/MMt/CNTOxi.

**Figure 2 polymers-13-02792-f002:**
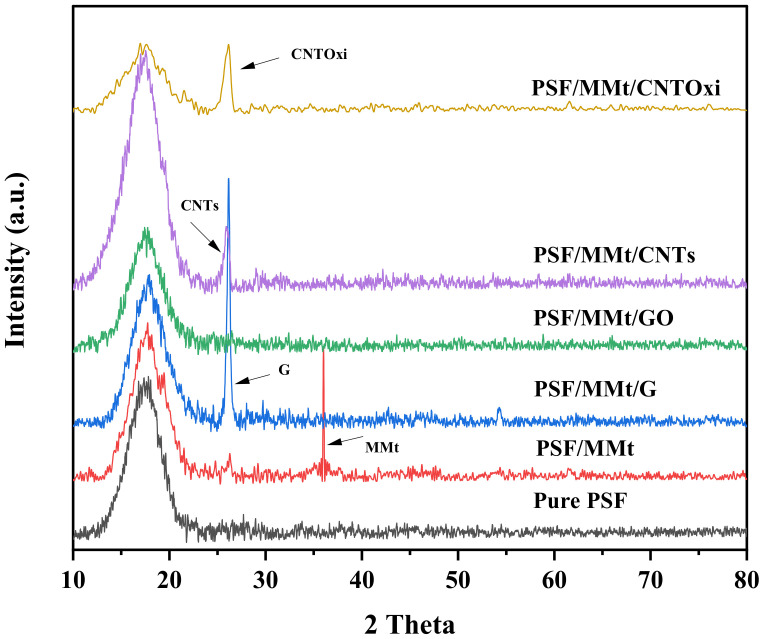
XRD patterns of pure PSF, PSF/MMt, PSF/MMt/G, PSF/MMt/GO, PSF/MMt/CNTs, and PSF/MMt/CNTOxi.

**Figure 3 polymers-13-02792-f003:**
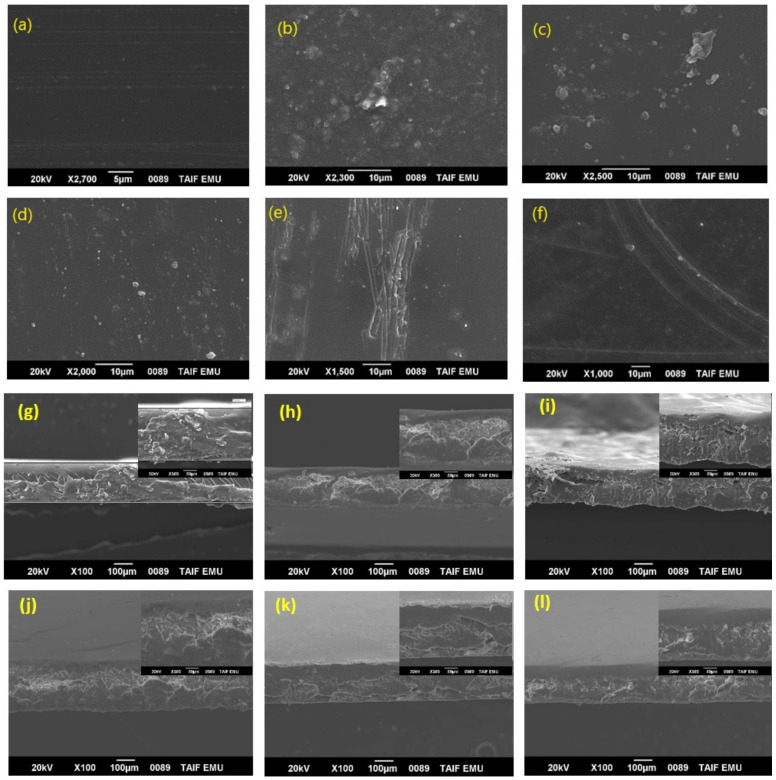
SEM images (top-surface and cross-section) of pure PSF (**a**,**g**), PSF/MMt (**b**,**h**), PSF/MMt/G (**c**,**i**), PSF/MMt/GO (**d**,**j**), PSF/MMt/CNTs (**e**,**k**), and PSF/MMt/CNTOxi (**f**,**l**).

**Figure 4 polymers-13-02792-f004:**
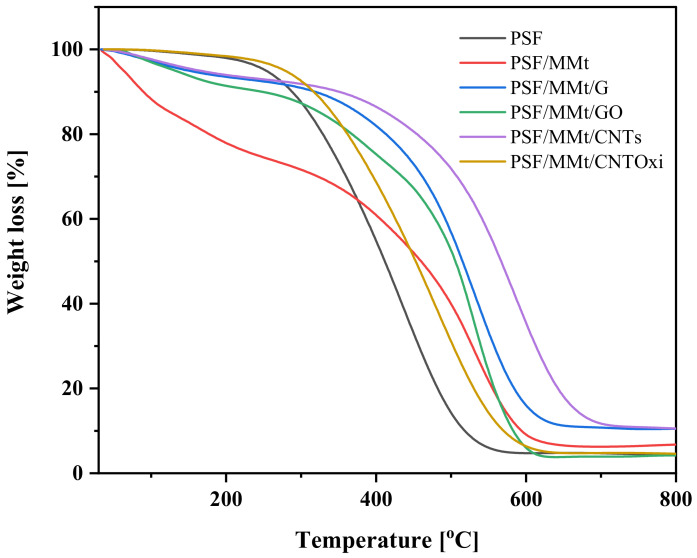
TGA curves of pure PSF, PSF/MMt, PSF/MMt/G, PSF/MMt/GO, PSF/MMt/CNTs, and PSF/MMt/CNTOxi.

**Figure 5 polymers-13-02792-f005:**
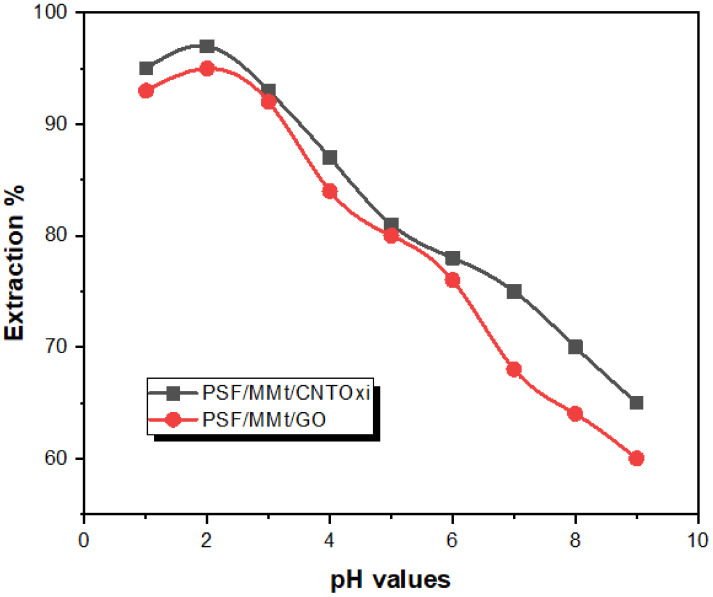
The pH effect on the adsorption of Hg^+2^ ion (2 mg/L) on 25 mg of PSF/MMt/GO and PSF/MMt/CNTOxi respectively at 25 °C.

**Figure 6 polymers-13-02792-f006:**
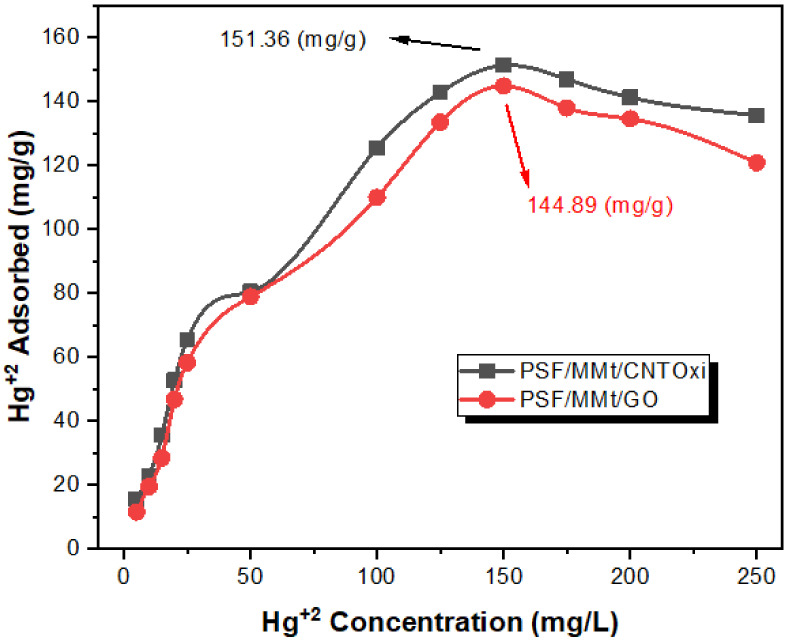
The adsorption profile of Hg^+2^ ion on 25 mg of PSF/MMt/GO and PSF/MMt/CNTOxi, respectively, in relation to the concentration at pH 2.0 and 25 °C.

**Figure 7 polymers-13-02792-f007:**
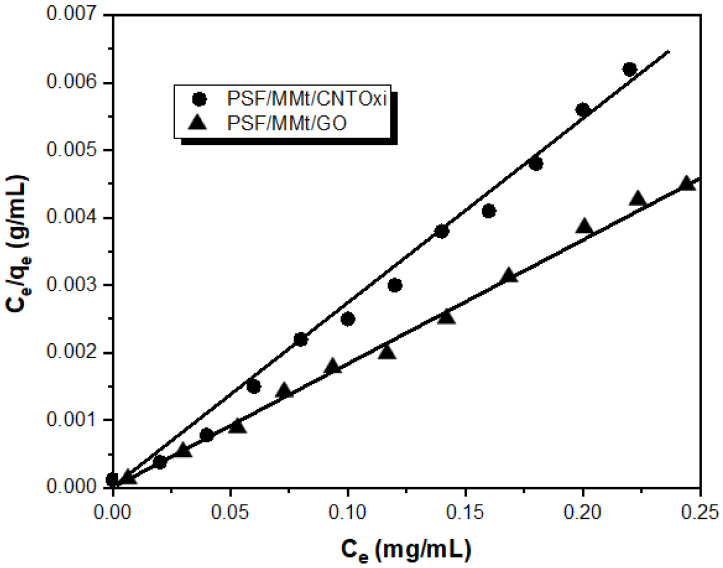
Langmuir adsorption isotherm model of Hg^+2^ ion adsorption on 25 mg of PSF/MMt/GO and PSF/MMt/CNTOxi respectively, at pH 2.0 and 25 °C. Adsorption experiments were obtained at different concentrations (0–250 mg/L) of Hg^+2^ under static conditions.

**Table 1 polymers-13-02792-t001:** Suggested list of abbreviations, codes and compositions for the fabricated materials.

Code	Clay % (g)	Carbon-Filler Loading (%)	Carbon-Filler Loading (g)
Pure PSF	-	-	-
PSF/MMt	5 (0.2)	-	-
PSF/MMt/G	2.5 (0.1)	2.5	0.1
PSF/MMt/GO	2.5 (0.1)	2.5	0.1
PSF/MMt/CNTs	2.5 (0.1)	2.5	0.1
PSF/MMt/CNTOxi	2.5 (0.1)	2.5	0.1

**Table 2 polymers-13-02792-t002:** Thermogravimetric data for the investigated samples.

	T_10_ (°C)	T_50_ (°C)	Residue at 750 (°C) %
Pure PSF	288	411	4.7
PSF/MMt	90	460	6.6
PSF/MMt/G	316	515	10.8
PSF/MMt/GO	246	505	4.7
PSF/MMt/CNTs	350	566	10.8
PSF/MMt/CNTOxi	316	452	4.7

**Table 3 polymers-13-02792-t003:** Flexural modulus of elasticity in tension and tensile strain at break of PSF and its related carbon-based nanocomposites.

Code	Flexural Properties			Modulus of Elasticity in Tension (MPa)	Tensile Strain at Break (%)
	Strength (MPa)	Modulus (GPa)	Toughness (kJ/m^2^)		
PSF	3.86 (±0.61)	96.68 (±0.28)	3.54 (±0.65)	73.15 (±1.6)	30.22 (±0.54)
PSF/MMt	4.47 (±0.98)	132.61 (±1.22)	4.03 (±0.23)	70.08 (±1.2)	29.43 (±0.27)
PSF/MMt/G	5.83 (±0.22)	134.92 (±1.07)	5.54 (±0.12)	54.21 (±1.5)	18.65 (±0.71)
PSF/MMt/GO	6.03 (±0.28)	139.32 (±1.02)	5.88 (±0.82)	58.27 (±1.7)	14.03(±0.56)
PSF/MMt/CNTs	5.46 (±0.12)	140.67 (±1.43)	4.98 (±0.75)	52.34 (±1.3)	16.05 (±0.32)
PSF/MMt/CNTOxi	6.84 (±0.16)	143.42 (±0.99)	5.15 (±0.41)	60.22 (±1.4)	12.94 (±0.19)

**Table 4 polymers-13-02792-t004:** Tensile properties and impact strength of PSF and its related carbon-based nanocomposites.

Code	Tensile Properties			Impact Strength (KJ/m^2^)
	Strength (MPa)	Modulus (GPa)	Toughness (kJ/m^2^)	
PSF	3.96 (±0.13)	99.26 (±0.87)	6.43 (±1.94)	1.25 (±0.86)
PSF/MMt	4.32 (±0.52)	114.57 (±1.08)	7.28 (±0.76)	1.45 (±1.18)
PSF/MMt/G	5.68 (±0.45)	128.28 (±1.25)	7.77 (±0.58)	1.60 (±1.04)
PSF/MMt/GO	6.07 (±0.86)	134.44 (±1.13)	9.34 (±1.11)	1.69 (±0.82)
PSF/MMt/CNTs	6.82 (±0.82)	139.92 (±1.05)	7.08 (±1.26)	1.62 (±1.07)
PSF/MMt/CNTOxi	7.54 (±0.91)	133.51 (±1.55)	8.40 (±1.31)	1.76 (±0.99)

**Table 5 polymers-13-02792-t005:** Surface selectivity study on the adsorption of pure PSF, PSF/MMt, PSF/MMt/G, PSF/MMt/CNTs, PSF/MMt/GO, and PSF/MMt/CNTOxi toward different metal ions at 25 °C.

Material	Metal Ions	*q_e_* (mg/g)	*K_d_* (mL/g)
PSF/MMt/CNTOxi	Hg^+2^	2.08	61,357.25
	Ni^+2^	0.11	49.57
	Fe^+3^	0.34	167.82
	Cr^+3^	0.23	32.41
	Y^+3^	0.37	45.67
	Al^+3^	0.46	147.69
	Pb^+2^	0.59	186.43
	Zn^+2^	0.14	55.67
	Co^+2^	0.29	42.83
	Sr^+2^	0.04	12.87
PSF/MMt/G	Hg^+2^	1.12	1829.46
PSF/MMt/GO	Hg^+2^	1.98	56,947.72
PSF/MMt/CNTs	Hg^+2^	1.19	1958.31
P/MMt	Hg^+2^	0.35	77.82
Pure PSF	Hg^+2^	0.17	62.17

**Table 6 polymers-13-02792-t006:** Parameters of Langmuir isotherm constants of PSF/MMt/CNTOxi and PSF/MMt/GO surfaces respectively against the adsorption of Hg^+2^, at pH 2.0 and 25 °C (N = 3).

Material	*Q_o_* (mg/g)	*b* (L/mg)	*R* ^2^	*R_L_*
PSF/MMt/CNTOxi	149.36	0.36	0.992	0.04
PSF/MMt/GO	142.48	0.28	0.985	0.02

## Data Availability

The data presented in this study are available on request from the author.

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
