# Peer review of "Polysulfone Membranes Based Hybrid Nanocomposites for the Adsorptive Removal of Hg(II) Ions"

_polymers, 2021, doi:10.3390/polym13162792_

Round 1
Reviewer 1 Report
Manuscript ID polymers-1333875
Title: Polysulfone Membranes Based Hybrid Nanocomposites for the Adsorptive Removal of Hg(II) ions
Comments:
Author presented “polysulfone membranes based hybrid nanocomposites for the adsorptive
removal of Hg(II) ions, Written of abstract is fine. Introduction contains up to date information,
however, author did not differentiate membrane type. In principle, membrane classically two
types: 1) filtration membrane and 2) adsorptive membrane. Author seem to deal with adsorptive
membrane due to basically a composite film is used in this study.
In experimental part, section 2.2. author used 5 mg of MWCNT for carboxylation. After acid
treatment how much yield no indication. Too little amount used to acid treatment resulted highly
percent error due to work-up procedure error. Secondly author used more than 5 mg sample of
MWCNT in the composite. Section 2.3 is incomplete description. Amounts of components are not
specified, just indicated MMt and graphene used 2.5% wt for each one. Some confusion came
from Table 1. For example, author listed 2.5 g PSF mixed with 0.2 g clay, by 0.2 g carbon filler
loaded gave 2.5% of carbon filler loading. Practically carbon filler loaded was few times more
than 2.5%. Need to rewrite this section, but no chance to alter all of parameters.
Section 2.4, author referred to ref. 7. In this reference 7 reported the batch adsorption process.
Author also needed a brief description of adsorption procedure, specifically amount of sorbent,
concentration of metal ions (mg/L) and amount of metal solution (mL).
In results and discussion part author presented a FTIR graph (Figure 1), arrow indicated very
small traceable peaks for C=O and OH groups due to GO and CNTOxi. There are so many such
traceable peaks around. I suggest to provided IR spectra of pure GO and acid treated CNTOxi
either embedded in Figure 1 or use supporting information.
TGA analysis Table 2 showed T10 (ï‚°C) and T50 (ï‚°C), what is relation with figure 4? Residue
at 750 (ï‚°C) % percent is meaningless because PST so early decreased and remaining 4.7%. About
580 (ï‚°C) is same 4.7% for PST. Need to write separately which temp. is constant %.
Figure 5 and Table 5 having same data, no informative Figure. Author may delete figure 5 or
send it to supporting information. Caption of Figure 6 needed all parameters of adoption
experiment. Figure 7 shows unusual behavior, because PSF/MMt/CNTOxi and PSF/MMt/GO
showed decreasing uptake capacity at higher concentration. Reported data showed that uptake
capacity either constant or very little increases at higher concentration after saturated the adsorbent
or adsorptive membrane/composites etc. No chemistry behind the reason of uptake capacity
decreased. Therefore, Langmuir isotherm data is plotted below 250 mg/L conc. It meant Langmuir
isotherm mode is limit at lower concentration.
Overall, I suggest to revise the manuscript according to comments provided before decide to
publish this work.
Reviewer 2 Report
This paper reports on Hg adsorption experiment on new polysulfone – clay – nanocarbon membranes. The study is in principle interesting, and can be published after eliminating some discrepancies.
Major remarks
- Check all the missing links in the in-text references
- The preparation of the composites is not clear. Please give the weights or weight fractions of all components, including PSF, for all composites in Table 1.
- The narrow diffraction peak of the MMt sample is very suspicious, please provide reference to support this, or check whether it is an artefact.
- In Conclusion, author writes: “ The thermal stabilities of the hybrid nanocomposite membranes were lower than that of pristine PSF“ However, TG results, fig 4, shows that the degradation of the pristine PSF happens at much lower temperature, then that of the composites. Please explain better this point.
Minor remarks
- A recent review can be added,
Graphene Oxide-Based Nanofiltration for Hg Removal from Wastewater: A Mini Review,
https://doi.org/10.3390/membranes11040269
- Please verify the X-ray wavelength, and the line of the characteristic radiation of the Cu target, on page 4.
- Please use fullstop as decimal separator, for example here: diffraction peak at 35,9°.
- Please follow the mdpi style for the references.
- Sentences that need improvements:
“ Then the resulting was washed by … “
“Then separately of MMt and …”
…”
Round 2
Reviewer 2 Report
In the revised version, Author clarified the problematic issues. Some minor improvements are still needed before the paper can be accepted to Polymers.
Minor corrections
This sentence can be improved:
... were fitted well by the Langmuir equation to described adsorption isotherm models, as illustrated in Fig. 7.
please use Palatino Linotype fonts throughout the text, according to the template. At some places, for example page 11 sentence: " Fig. 6 also shows that there was a small decrease" ,another type is used.
section 2.3. sentence: the samples composition is shown in xxxx -- please use here a plain number instead of field link.
section 3.1.1. sentence: were collected from 4000 to 400 cm-1, as shown in xxxxx- please use here a plain reference number.
section 2.5, the proper Angstrom symbol should be used. MS word insert special characters.
section 3.1.3., should be written Fig. 3g-l
In the paragraph between equations 2 and 3, Qo and Ce and others should be written using lower indices, similar to the notations used in the equations.
In a few citations, the page numbers are missing, e.g. 9, 33, 35